# Unlocking the Power of GANs in Non-Autoregressive Text Generation under Weak Conditions

## Abstract

Non-autoregressive (NAR) models once attracted significant attention from the research community but have received considerably less focus in the pursuit of general artificial intelligence. Our analysis reveals that the convergence problem in existing fully NAR models trained under Maximum Likelihood Estimation (MLE) becomes more severe in tasks where the input does not provide definitive semantic constraints for the output. We denote these input conditions as weak conditions, which encompass most "creative" tasks. Consequently, existing fully NAR models struggle to achieve satisfactory performance in such scenarios and remain confined to limited application domains. This limitation hinders fully NAR models from keeping pace with the rapidly evolving demands of diverse and challenging tasks. Unlike MLE, which is fundamentally incompatible with NAR models, Generative Adversarial Networks (GANs) offer superior theoretical convergence guarantees and inference characteristics for fully NAR architectures. We therefore propose an Adversarial Non-autoregressive Transformer (ANT) based on GANs specifically designed for weak condition tasks. ANT incorporates two key innovations: 1) Position-Aware Self-Modulation to provide more effective input signals, and 2) Dependency Feed Forward Network to enhance dependency modeling capabilities. Experimental results demonstrate that ANT achieves comparable performance to mainstream models while maintaining significantly higher efficiency, and exhibits substantial potential in various applications including latent interpolation and semi-supervised learning.

## 1 Introduction

Non-autoregressive (NAR) models, known for their lower decoding latency compared to autoregressive (AR) models, once attracted significant attention from the research community (Huang et al., 2022a). Among them, fully NAR models require only one step to obtain samples, thereby maximizing the speedup of this model family. Nevertheless, these models have diminished in prominence during the era of Large Language Models (LLMs), despite their potential for addressing the challenge of high decoding latency in LLMs. The main obstacle stems from the convergence problem inherent in existing fully NAR models. These models are trained using Maximum Likelihood Estimation (MLE); however, MLE-based fully NAR models maintain non-negative lower bounds between the learned distributions and target distributions (Huang et al., 2022a). In other words, the learned distributions cannot be identical to the target distributions unless all words in a sentence are independent of each other—a condition that does not reflect real-world scenarios. This ultimately leads to the multi-modality problem (Gu et al., 2018), where these models tend to mix words from different candidates and produce ungrammatical results.

This inherent problem directly limits the extension of fully NAR models to various tasks. Existing fully NAR models are primarily developed for several specific tasks such as machine translation (Gu et al., 2018) and text summarization (Liu et al., 2022). The input in these tasks provides and constrains the definite semantic meaning of the output, with one or several output candidates representing this same meaning. We refer to the condition of such tasks as a **strong condition**. An example of a classical task with strong conditions, machine translation, is shown in Figure 1 (a). These tasks always exhibit clear corresponding relations between input and output, and require only several, or even one acceptable result. Existing fully

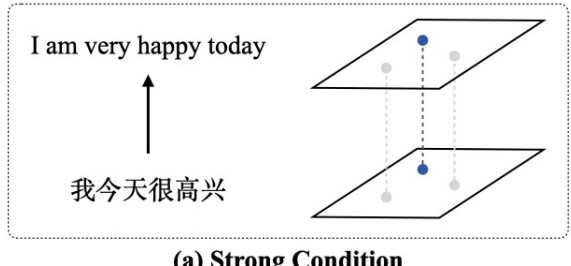 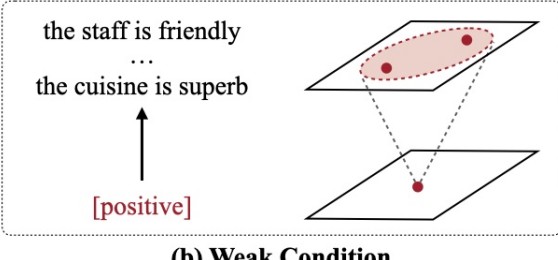

(a) Strong Condition     (b) Weak Condition

Figure 1: Comparison between strong conditions and weak conditions. (a) Translating a sentence. (b) Generating comments based on an emotion label. Weak condition inputs always have diverse and more candidates.

NAR models can incorporate element-level (e.g., words) mapping relations (Gu et al., 2018) or simplify target distributions (Kim & Rush, 2016) to hide the inherent problem and obtain satisfactory performance in tasks with strong conditions (Xiao et al., 2022).

However, there is another category of tasks where the input does not constrain the semantic meaning of the output, which are denoted as tasks with **weak conditions**. These tasks include most "creative" tasks such as sentence completion (Brown et al., 2020), story generation (Fan et al., 2018; Guan et al., 2021), and poetry creation (Ormazabal et al., 2022). As shown in Figure 1 (b), an input condition in these tasks describes a region in the output space, and all the samples in this region are output candidates. Unlike strong condition tasks, tasks with weak conditions do not have strong corresponding relations between input and output, and they require models to learn the original target distributions instead of simplified ones. The multi-modality problem inherent in existing MLE-based fully NAR models can no longer be hidden. Even worse, there are always more candidates in weak condition tasks, so the problem becomes more severe, and models will mix more unrelated words to obtain meaningless samples. Existing NAR models thus struggle to achieve satisfactory performance in tasks with weak conditions, and their development has stagnated in strong condition tasks. The limited application scenarios of existing NAR models have led to a decline in attention towards them in the quest for general artificial intelligence.

The inability of MLE-based fully NAR models to handle tasks with weak conditions motivates us to consider another family of generative models: Generative Adversarial Networks (GANs) (Goodfellow et al., 2014). The synthetic distributions learned by GANs can theoretically converge to the real distributions, ensuring their convergence even under weak conditions. Moreover, GANs can generate high-quality samples in a single forward pass, which precisely aligns with the requirements of NAR models. Rather than adopting unstable *REINFORCE* (Williams, 1992) or biased *continuous relaxations* (Jang et al., 2017) to address the non-differentiable sampling operation in language GANs, we follow the research line of representation modeling methods (Ren & Li, 2024) and propose an **Adversarial Non-autoregressive Transformer (ANT)** for weak condition tasks.

ANT incorporates two key features: Position-Aware Self-modulation for generating clear signals that enable the model to produce diverse words within a sentence; and Dependency Feed Forward Network (Dependency FFN) for helping the model capture more accurate word dependencies during the unstable training process of GANs. The experimental results demonstrate that ANT significantly narrows the performance gap between AR and fully NAR models in tasks with weak conditions, while maintaining high inference efficiency. The contributions of this paper can be summarized as follows:

- We reveal the limitations of existing (MLE-based) NAR models in "creative" tasks by analyzing their constraints under weak conditions from both empirical and theoretical perspectives. Furthermore, we demonstrate that GANs represent a more suitable approach for constructing fully NAR models, as their convergence is guaranteed and their highly efficient inference capabilities are well-suited for NAR models.

- Based on GANs, we propose an Adversarial Non-autoregressive Transformer (ANT). ANT incorporates two key features: 1) Position-Aware Self-modulation, which provides more effective input signals to assist the model in generating diverse words within a sentence; and 2) Dependency Feed Forward Network (Dependency FFN), which further improves model performance by enhancing its capacity for dependency modeling.

- We compare the performance of ANT with existing models in tasks with weak conditions. The experimental results demonstrate that ANT achieves comparable performance to other models with significantly lower decoding latency. We further demonstrate the substantial potential of ANT in various applications such as latent interpolation and semi-supervised learning. To the best of our knowledge, this is the first work to demonstrate the effectiveness of GANs in building fully NAR models.

The rest of this paper is organized as follows: Section 2 introduces the background and limitations of existing NAR models in weak condition tasks. Section 3 presents the details of ANT. Section 4 describes the experimental settings and results. Finally, we draw out conclusion in Section 5.

## 2 Background

### 2.1 Limitations of MLE in Weak Condition Tasks

Most existing fully NAR models are based on MLE and have low decoding latency, but always sacrifice sample quality. They tend to mix words from different candidates due to the lack of word dependencies. This problem, which is knows as the multi-modality problem, will be augmented in weak condition tasks. Weak conditions describe a region in the semantic space of output which means there are more diverse candidates and the models will mix more unrelated words together in a sentence. This problem can also be illustrated from a theoretical perspective.

#### 2.1.1 Theoretical Analysis

Given a MLE-based NAR model $P_\theta(Y|X)$, Huang et al. (2022a) reveal a non-negative lower bound between the leaned distribution and the real distribution. More specifically, $min_\theta KL(P_{data}(Y|X)||P_\theta(Y|X)) \geq C$, where $C = \sum_{i=1}^{l} H_{data}(y_i|X) - H_{data}(Y|X)$, and $H_{data}(\cdot|X)$ is the Shannon Entropy. $C$ is also known as conditional total correlation. Based on it, we further analyze the difference of the conditional total correlation when the scenario is changed from strong conditions to weak conditions. We analyze this problem by considering the transformations from two different input $\mathbb{A}$ and $\mathbb{B}$ to output $\mathbb{T}$. Due to the complexity of real data, our analysis is based on the simplifications in **Assumption 1**.

**Assumption 1.** *Each target sequence $Y \in \mathbb{T}$ consists of $l$ elements. There are no identical elements in the candidates. Each input $X_A \in \mathbb{A}$ has $n$ target sequences as candidates and each input $X_B \in \mathbb{B}$ has $m$ target candidates. Each $Y$ can only be mapped by a specific input. All the input and target sequences follow uniform distributions.*

Based on **Assumption 1**, we obtain **Theorem 1** to describe the difference of the conditional total correlation between two different input $\mathbb{A}$ and $\mathbb{B}$.

**Theorem 1.** *The difference between the conditional total correlation of input $\mathbb{A}$ and $\mathbb{B}$ has the following relation: $C_A - C_B = (l-1) \cdot log(n/m)$.*

The proof can be found in Appendix A. **Theorem 1** reveals the relations between the difference of the conditional total correlation, sequence length $l$, and candidate numbers $n$ and $m$. We further obtain a mark on **Theorem 1**:

**Mark 1.** *If $n > m$, then $C_A > C_B$. Higher $l$ and $n$ lead to higher difference between $C_A$ and $C_B$.*

Comparing to the strong conditions, input in weak conditions always corresponds to more possible candidates, so its conditional total correlation will be higher. The larger gap between the learned distributions and the real distributions will lead MLE-based NAR models to generate more ungrammatical results in weak conditions. Unlike MLE-based NAR models whose convergence cannot be guaranteed and is easily influenced by input conditions, the convergence of GANs has been proven regardless of input conditions Goodfellow et al. (2014). Thus, it denotes a more promising method to train fully NAR models under weak conditions.

### 2.1.2 Existing NAR models in Tasks With Weak Conditions

Many techniques in existing NAR models can be unified in a framework (Huang et al., 2022a): enhancing inputs (Ghazvininejad et al., 2019) or modifying targets (Gu et al., 2018; Du et al., 2021). Among them, there are two most popular techniques: 1) simplifying output with knowledge distillation (Kim & Rush, 2016); 2) enhancing input based on multiple iterations (Ghazvininejad et al., 2019). Different with tasks in strong conditions only requiring several acceptable results, weak condition tasks require models to obtain diverse results. This process describes a region in the output space. Using techniques like knowledge distillation to obtain a simplified ones is no longer acceptable. Besides, models like conditional masked language model (CMLM) (Ghazvininejad et al., 2019) and diffusion-lm (Li et al., 2022) uses multiple inference steps to improve performance, but their efficiency is sacrificed. The inapplicability of these two techniques renders the vast majority of NAR models unsuitable for tasks with weak conditions, since most of them are deeply bound to at least one of them.

In addition, researchers attempt to address the multi-modality problem by proposing alternative modeling methods that do not rely on the aforementioned techniques. DA-Transformer (Huang et al., 2022b) is a classical one. However, it is still based on MLE and its convergence has not yet been theoretically proved. When we migrate it to weak condition tasks, it obtains meaningless samples and fail to converge. It again demonstrates the high reliance of existing NAR models to strong conditions, and the challenges of weak condition tasks to existing MLE-based NAR models.

### 2.2 Language GANs

Different with the incapacity of MLE in building NAR models for weak condition tasks, GANs (Goodfellow et al., 2014) indicate a more promising training method. Their global optimal is achieved if and only if the learned distribution is exactly the same as the target one (Goodfellow et al., 2014). Unlike other popular generative models (e.g., autoregressive models (Yu et al., 2022) and diffusion models (Ho et al., 2020)), GANs can produce high quality samples in a single forward pass, which exactly meets the needs of building fully NAR models.

However, most of existing language GANs are based on AR structures (Yu et al., 2017; Che et al., 2017; Lin et al., 2017; Fedus et al., 2018), and lost the high efficiency nature of GANs. To tackle the non-differentiable sampling operation, they adopt *REINFORCE* (Williams, 1992) or *continuous relaxations* (Jang et al., 2017). REINFORCE methods use the discriminator to calculate rewards and guide the generator to updates its parameters based on the reward. However, this methods have been proven to be unstable and suffer from high variance (Lin et al., 2020). This would exacerbate the inherent instability of the GAN training process. Continuous relaxations methods, on the other hand, use a continuous distribution to approximate the discrete one. However, it is also proven that these methods are biased estimators (Lin et al., 2020). Both of them are not suitable for building GAN-based NAR models

Our work thus follows another research line, namely representation modeling method (Ren & Li, 2024). This method firstly maps words into representations and trains a generator to model these representations. It avoids the non-differentiable sampling operation during training process, so the gradients from discriminators can pass through to generators directly. This method is free from the limitations in *REINFORCE* or *continuous relaxations* and is demonstrated to be an effective way to train language GANs (Ren & Li, 2024).

NAGAN (Huang et al., 2021), which also adopts GANs to build NAR models, is the most similar work to ours, yet our work is different in three aspects: 1) we give in-depth analyses about the limitations of MLE-based NAR models from the perspective of weak conditions; 2) the performance of their model is significantly

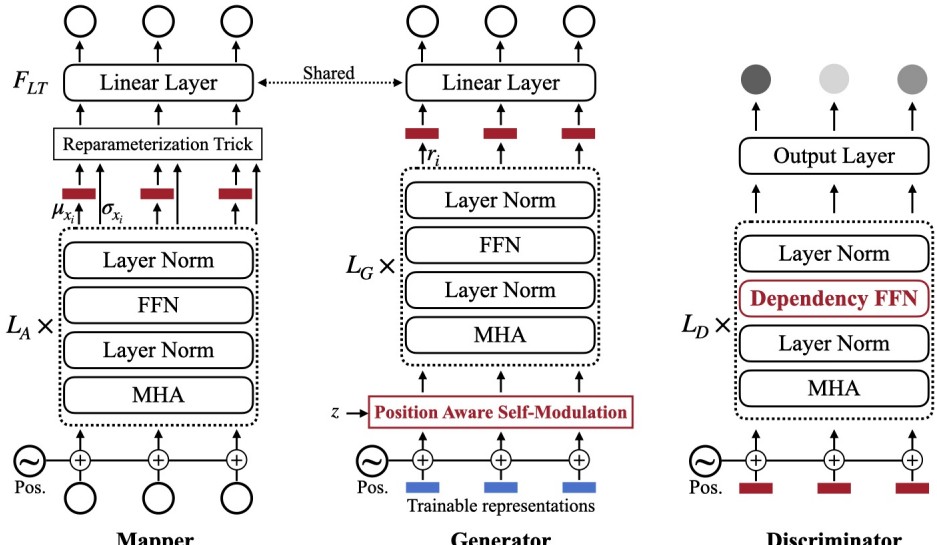

Figure 2: Structure of Adversarial Non-autoregressive Transformer (ANT). There are three modules: Mapper, which maps words into representations; Generator, which obtains word representations starting from random noises; and Discriminator, which identifies whether input representations are from Mapper or Generator. Position-Aware Self-Modulation is adopted in Generator to provide more effective signal, while Dependency FFN in Discriminator is used to model more accurate word dependencies.

limited by the biased *straight-through estimator* (Bengio et al., 2013), whereas our model is free from this problem; 3) our proposed facilities: Position-Aware Self-Modulation and Dependency Feed Forward Network can further boost model performance and they are not explored in previous work.

## 3 Model

### 3.1 Model Structure

In this paper, we propose an Adversarial Non-autoregressive Transformer (ANT) which generates text in a fully NAR manner. ANT is based on the representation modeling framework (Ren & Li, 2024). As shown in Figure 2, there are three parts in ANT: Mapper, Discriminator and Generator. The mapper converts words into representations, and the generator tries to recover these representations. The discriminator needs to identify whether input representations are from the mapper or the generator. We adopt Transformer (Vaswani et al., 2017) as the backbones of all the three parts to support highly parallel computation.

$$
\begin{aligned}
h_0 &= h_0' + E_{pos} \\
h_i' &= LN(MHA(h_{i-1}) + h_{i-1}) \\
h_i &= LN(FFN(h_i') + h_i')
\end{aligned}
\tag{1}
$$

An input is firstly added with a positional encoding and fed into encoder layers. Each encoder layer has a multi-head attention (MHA) module and feed forward network (FFN) module. A layer normalization is added after each module.

Inspired by BERT (Devlin et al., 2019), the mapper is trained to reconstruct words based on the masked input. However, instead of using cross entropy as the loss function, we follow the previous work which adopts representation modeling methods to train language GANs (Ren & Li, 2024), and use the loss function of variational autoencoder (VAE) (Kingma & Welling, 2014) to train the mapper:

$$
L_M = -\mathbb{E}_{z_i' \sim q(z_i'|x_i)}(log p(x_i|z_i')) + KL(q(z_i'|x_i)||p(z_i'))
\tag{2}
$$

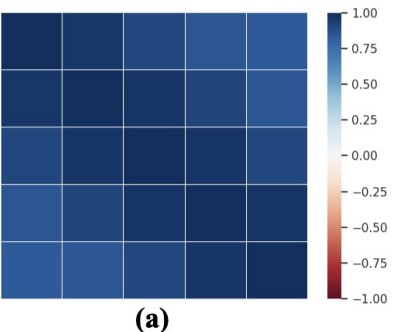 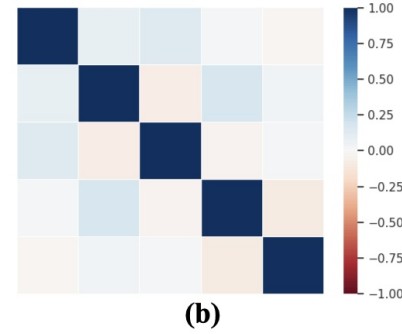

**(a)** **(b)**

Figure 3: Cosine similarity of the output from (a) Self-Modulation and (b) Position-Aware Self-Modulation. The representations obtained by Self-Modulation exhibit high similarity, while the representations obtained by Position-Aware Self-Modulation are highly diverse across different positions.

where $x_i$ is the $i$-th word in the sentence, $q(z_i'|x_i)$ is the process of encoded each words into latent variables $z_i'$, $p(x_i|z_i')$ is the process of recovering words based on $z_i'$. The latent variable $z_i'$ is obtained by using reparameterization trick: $z_i' = \mu_{x_i} + \sigma_{x_i} \cdot \mathcal{N}(0,1)$, and it is transformed back into words with a linear transformation layer $F_{LT}$. Different from cross entropy which maps words into specific points in the representation space, this method describes a region for each word, so representations slightly away from their central points $\mu_{x_i}$ can still be transformed into correct words (Ren & Li, 2024).

A non-autoregressive generator cannot input previously generated words, so trainable representations are adopted as input. The generator then gives output representations $r_i$ in different positions and uses the same linear transformation layer $F_{LT}$ in the mapper to transform these representations back into words. The discriminator adopts the output representations from the mapper and the generator ($\mu_{x_i}$ and $r_i$) as input. Different from image GANs whose discriminators give a single scaler output for an image, our discriminator gives output for each representation. During training, the mapper will be trained first, and its parameters are fixed during the training of the discriminator and the generator. The representations given by the generator need not be transformed into words in training process, so the gradients from the discriminator can directly pass through to the generator.

Causal masks are adopted in both the discriminator and the generator to break the possible symmetry in the input. We use Wasserstein distance (Arjovsky et al., 2017) as the training objective and adopt Lipschitz penalty (Petzka et al., 2018) to regularize the discriminator. However, there is still a gap between our basic model and existing autoregressive models and we further propose Position-Aware Self-Modulation and Dependency Feed Forward Network (Dependency FFN) to improve model performance.

### 3.2 Position-Aware Self-Modulation

An effective sampling method plays a key role in the success of GANs. The sampling operations in GANs are performed by incorporating latent variables, which are typically sampled from a standard normal distribution, to inject stochasticity and enable diverse output generation. Transformer-based image GANs (Lee et al., 2021) accomplish this process by adopting self-modulation (Chen et al., 2019). As shown in Figure 4 (a), this method starts from a trainable input matrix, which is then normalized with layer normalization and incorporates latent variables by calculating scale and shift vectors for the matrix. The scale and shift vectors modulate the normalized features, allowing the latent variables to influence the generation process. The results are then fed into the Transformer block for further processing to ultimately recover the word representations.

However, self-modulation assigns the same shift and scale vectors to the normalized results across different positions, which causes the input representations at various positions to be highly similar even with positional encodings (as shown in Figure 3 (a)). In contrast, the output of the generator (i.e., word representations in different positions) exhibits high diversity, as different words naturally occupy different regions in the

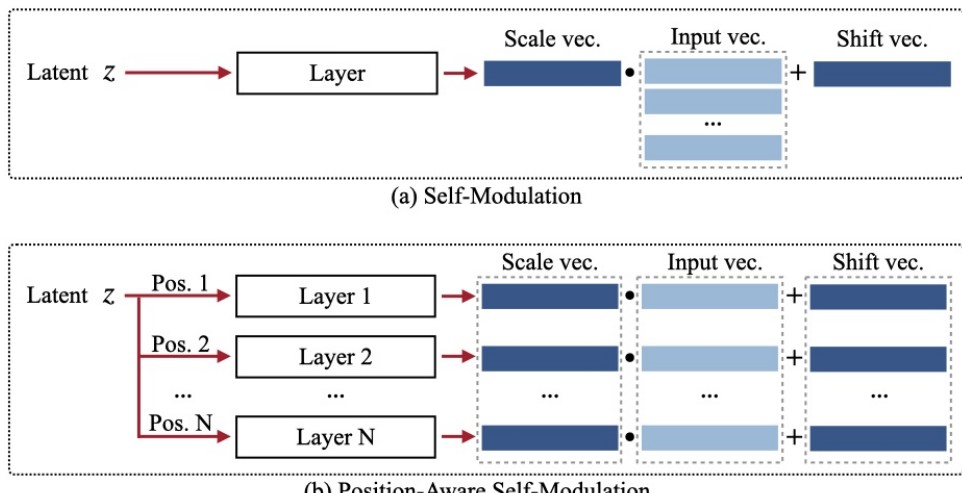

Figure 4: Comparison between (a) Self-Modulation and (b) Position-Aware Self-Modulation. Self-Modulation uses the same scale and shift vectors to manipulate input representations across all positions, while Position-Aware Self-Modulation adopts unique layers for different positions to calculate position-specific scale and shift vectors.

representation space. Such similar input representations cannot adequately capture the diversity among different words, thus leading to inaccurate sentence generation. This mismatch between uniform inputs and diverse outputs creates a bottleneck that limits the model's ability to generate high-quality, diverse text sequences.

To tackle this problem, we propose **Position-Aware Self-Modulation**. As shown in Figure 4 (b), this method adopts different mapping layers for the calculations in different positions so as to gain diverse results. In practice, Position-Aware Self-Modulation also starts from a trainable matrix, but it adopts different layers for different positions. To ensure high efficiency, this process is implemented a parallel manner, which is:

$$
\begin{pmatrix} \mathbf{h}'_1 \\ \mathbf{h}'_2 \\ \vdots \\ \mathbf{h}'_N \end{pmatrix} = MLP(z) \tag{3}
$$
$$
\mathbf{h}_i = \gamma(\mathbf{h}'_i) \circ LN(\mathbf{x}_i) + \beta(\mathbf{h}'_i)
$$

where $z$ is the latent variable (which is sampled from a standard normal distribution in this work), $\mathbf{x}_i$ is the $i$-th vector in the trainable input matrix, $\mathbf{h}'_i$ is the hidden representation in the $i$-th position, $MLP(\cdot)$ is a non-linear transformation whose activation function is GELU (Hendrycks & Gimpel, 2016), $LN(\cdot)$ is the layer normalization, $N$ is the length of the sentence, and $\gamma(\cdot)$ and $\beta(\cdot)$ are linear transformations. In Position-Aware Self-Modulation, representations in different positions are calculated based on unique parameters and have clear differences (as shown in Figure 3 (b)), so as to provide more effective signals to obtain target sentences.

### 3.3 Dependency Feed Forward Network

Transformer (Vaswani et al., 2017) builds word dependencies by dynamically assigning weights in the attention mechanism, where the self-attention mechanism computes pairwise relationships between positions in a sequence to capture long-range dependencies. This process, however, becomes unstable under the adversarial training dynamics of GANs, where the generator and discriminator are engaged in a minimax optimization that can lead to oscillatory behavior and training instability. During adversarial training, the competing objectives between generator and discriminator can interfere with the learning of attention patterns, making

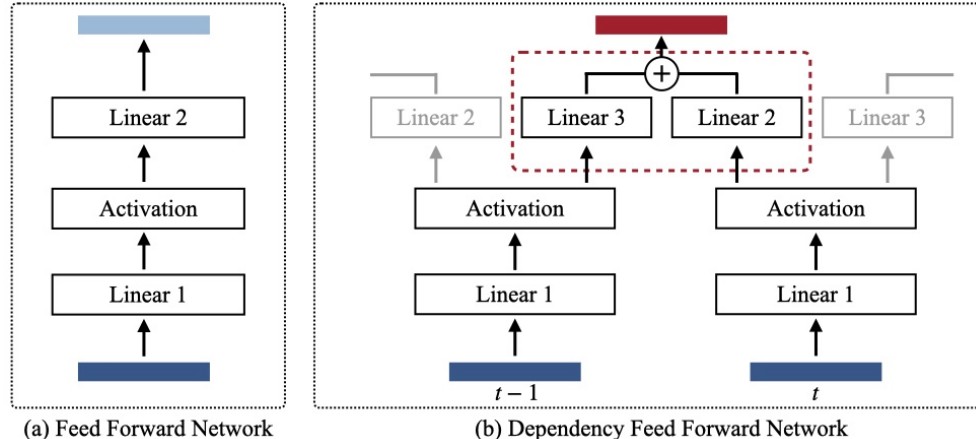

Figure 5: Comparison between (a) Feed Forward Network (FFN) and (b) Dependency Feed Forward Network (Dependency FFN). FFN processes the hidden representations at different positions independently, while Dependency FFN explicitly builds connections between the representations of different timesteps.

it difficult for the model to consistently capture syntactic and semantic dependencies between words. Consequently, this instability in dependency modeling can result in the generation of ungrammatical sentences that lack coherent structure and semantic consistency.

To address this limitation, we propose the Dependency Feed Forward Network (Dependency FFN), which explicitly strengthens the traditional FFN module with enhanced capacity for dependency modeling. Unlike the standard FFN that processes each position independently (as shown in Figure 5 (a)), our Dependency FFN incorporates inter-word relationships directly into the feed-forward computation. The structure of Dependency FFN is shown in Figure 5 (b), and calculated as follows:

$$
\begin{aligned}
\mathbf{s}_t &= \sigma(\mathbf{x}_t W_s + b_s) \\
\mathbf{o}_t &= \mathbf{s}_{t-1} W_a + \mathbf{s}_t W_b + b_o
\end{aligned}
\tag{4}
$$

where $\sigma(\cdot)$ is an activation function which is GELU in this work. With causal masks, $\mathbf{s}_{t-1}$ and $\mathbf{s}_t$ contain the information of first $(t-1)$ and $t$ words, respectively. Using the sum of these two variables can help the model to explicitly build stable dependencies between the $t$-th word and previous $(t-1)$ words in the fragile training process of GANs.

### 3.4 Extension to Conditional Generation

Besides unconditional generation, conditional generation is frequently employed in a variety of tasks. We thus also extend ANT to conditional generation. Given a condition representation $c$, the generator can consider it by shifting the original latent variable $z$. We find that using trainable factors to assign weights to $z$ and $c$ can slightly improve the performance. Thus, we incorporate the condition as follows:

$$
\hat{z} = \alpha_1 \circ z + \alpha_2 \circ c
\tag{5}
$$

where $\alpha_1$ and $\alpha_2$ are two trainable variables. Then, $\hat{z}$ is fed into Position-Aware Self-Modulation, so the generator can consider the condition representation.

For the discriminator, we use the sum of word representations $x_t^d$ and conditional representations $c$ as the input: $\hat{x}_t^d = x_t^d + c$. Then, $\hat{x}_t^d$ is fed into the remaining modules of the discriminator.

Table 1: FED and I. BLEU on the COCO Dataset and EMNLP Dataset.

| Model | DI | COCO | | EMNLP | |
|---|---|---|---|---|---|
| | | FED ↓ | I. BLEU ↑ | FED ↓ | I. BLEU ↑ |
| Training Data | - | 0.007 | 35.36 | 0.010 | 20.62 |
| Transformer | O(N) | **0.008** | **34.28** | **0.014** | **19.50** |
| RelGAN | O(N) | 0.062 | 29.53 | 0.136 | 14.74 |
| ScratchGAN | O(N) | 0.014 | 30.76 | 0.018 | 17.19 |
| InitialGAN | O(N) | 0.013 | 33.06 | 0.025 | 17.74 |
| CMLM | O(k) | 0.016 | 27.65 | 0.062 | **16.67** |
| NAT | O(1) | 0.024 | 26.41 | 0.111 | 11.38 |
| NAGAN | O(1) | 0.084 | 24.98 | 0.748 | 2.01 |
| ANT | O(1) | **0.013** | **31.12** | **0.026** | 15.51 |

## 4 Experiment

### 4.1 Experiment Setup

The experiment focuses on tasks with weak conditions and covers both unconditional generation and conditional generation to evaluate model performance comprehensively. For the unconditional generation, the task is to generate sentences whose distribution can be as close as to the target sets. We follow the settings of previous work (de Masson d'Autume et al., 2019; Ren & Li, 2024) and use sentences from two datasets: the COCO Image Caption Dataset (Lin et al., 2014)[1] and the EMNLP 2017 News Dataset[2]. The size of training sets of the COCO dataset and the EMNLP dataset are set to be 50,000 and 200,000, respectively. The COCO dataset can support evaluations in short sentence generation, while the EMNLP dataset focuses on long sentence generation. For the conditional generation, we randomly select 100,000 sentences from the Yelp Dataset[3] as training data and use emotion labels (positive or negative) as conditions.

### 4.2 Evaluation Metrics

The evaluation is conducted at both embedding level and token level. In embedding level, we use Universal Sentence Encoder[4] (Cer et al., 2018) to transform sentences into embeddings. Then, we calculate both **Fréchet Embedding Distance (FED)** (de Masson d'Autume et al., 2019) and **Least Coverage Rate (LCR)** (Ren & Li, 2024) to evaluate the overall similarity and the fine-grained similarity of two distributions, respectively.

In token level, we use **Inverse-BLEU (I. BLEU)** to evaluate model performance in terms of quality and diversity together. Besides, we also draw a curve of **BLEU** (Papineni et al., 2002) and **Self-BLEU** (Zhu et al., 2018) by tuning the temperature of the model (Caccia et al., 2020). In the case of conditional generation, **Accuracy (Acc.)** is also employed to assess whether the models produce sentences that align with the input labels.

### 4.3 Compared Model

An important compared model is Transformer, which adopts AR structures and is trained on MLE. It is the mainstream model in various text generation tasks. In addition, we also compare the performance between our models and existing AR language GANs: SeqGAN (Yu et al., 2017), RankGAN (Lin et al., 2017), MaliGAN (Che et al., 2017), LeakGAN (Guo et al., 2018), and ScratchGAN (de Masson d'Autume et al., 2019), which are based on *REINFORCE*; RelGAN (Nie et al., 2019), which uses *Gumbel-softmax* to obtain gradients; InitialGAN (Ren & Li, 2024), which does not use the above two method and adopts representation

---

[1]https://cocodataset.org
[2]http://www.statmt.org/wmt17/
[3]https://www.yelp.com/dataset
[4]https://tfhub.dev/google/universal-sentence-encoder/4

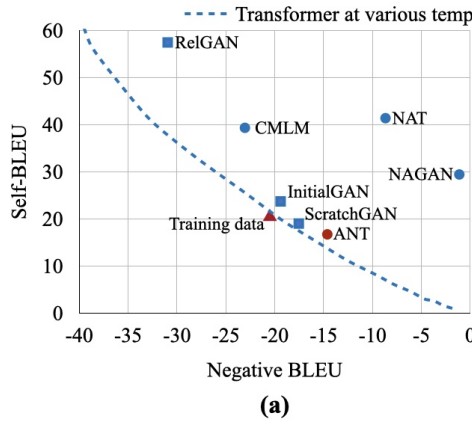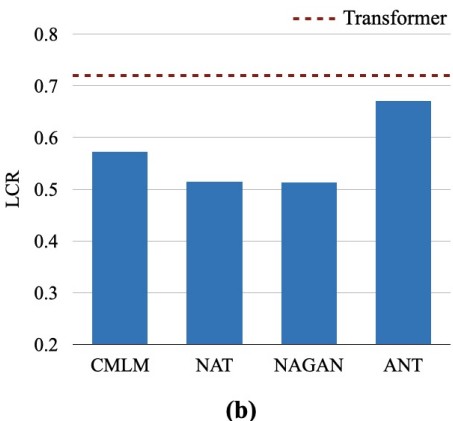

(a)          (b)

Figure 6: Additional Experimental Results: (a) Negative BLEU and Self-BLEU curve on the EMNLP dataset. Lower is better; (b) Least Coverage Rate (LCR) on the COCO dataset. Higher is better.

Table 2: FED, I. BLEU and Acc. on the Yelp dataset

| Model | DI | FED | I. BLEU | Acc. |
|---|---|---|---|---|
| Training Data | - | 0.008 | 24.18 | 92.47% |
| Transformer | O(N) | 0.011 | 23.04 | 91.73% |
| CMLM | O(k) | **0.015** | 18.35 | 87.85% |
| NAT | O(1) | 0.032 | 11.81 | 83.54% |
| ANT | O(1) | 0.018 | **19.08** | **88.35%** |

modeling. All the models mentioned-above are AR models whose Decoding Iteration (DI) is $O(N)$ ($N$ is the sequence length).

For NAR models, we compare with another GAN-based model: NAGAN (Huang et al., 2021). Furthermore, two classical and popular MLE-based NAR structures are compared in our experiments: Non-autoregressive Transformer (NAT) (Gu et al., 2018) and conditional masked language model (CMLM) (Ghazvininejad et al., 2019). More illustrations about experiment details can be found in Appendix A. We will release our code to the public in the future.

### 4.4 Implementation Details

The layer numbers of the mapper, generator and discriminator are all set to be 4. Their input dimension is 256, and the hidden dimension of FFN / Dependency FFN is 1,024. The head number is set to be 8. We use AdamW (Loshchilov & Hutter, 2019) as the optimizer of the mapper and the weight decay is set to be 1e-5; its learning rate is 0.0001. During the adversarial training, AdamW (Loshchilov & Hutter, 2019) is used as the optimizer of the discriminator whose weight decay is set to be 0.0001; its learning rate is 0.0002 for the COCO and Yelp dataset, and 0.00015 for the EMNLP dataset. We choose Adam (Kingma & Ba, 2015) as the optimizer of the generator and its learning rate is 0.0001. The $\beta_1$ and $\beta_2$ in the optimizers of the discriminator and the generator are set to be 0.5 and 0.9, respectively. The maximum training epoch is set to be 4,500. We implement our model based on Tensorflow[5] (Abadi et al., 2015) and the model is trained on NVIDIA GeForce RTX 3090.

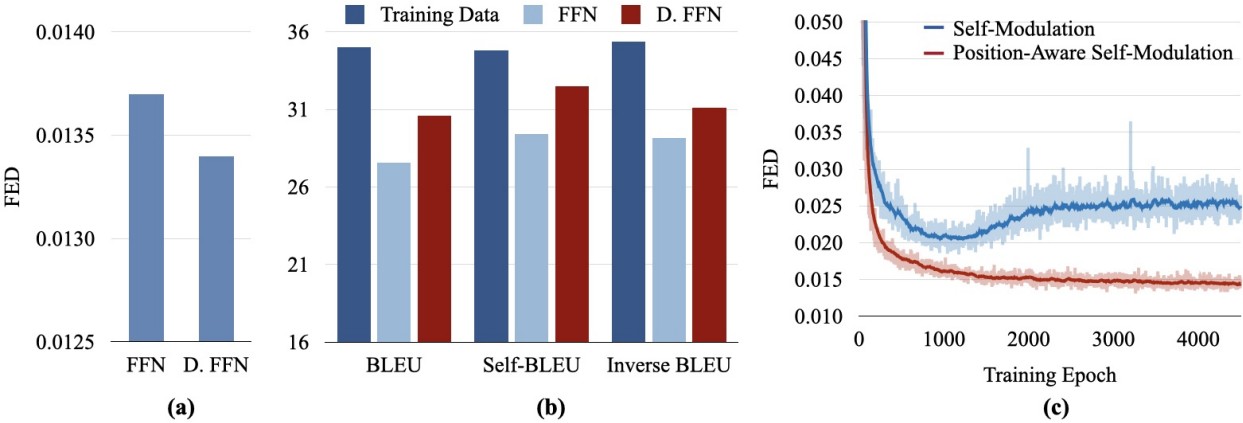

Figure 7: Ablation study of our proposed components. (a) Comparison of FED between FFN and Dependency FFN (D. FFN). (b) Comparison of BLEU, Self-BLEU, and Inverse BLEU metrics between FFN and Dependency FFN. (c) Training curves comparing self-modulation with Position-Aware Self-Modulation.

## 4.5 Experimental Result

### 4.5.1 Unconditional Generation

The experimental results of the unconditional generation are shown in Table 1. "DI" indicates Decoding Iteration. For the COCO dataset, Transformer performs best among all compared models, and there are large gaps between MLE-based NAR models (NAT and CMLM) and AR models, which is consistent with our theoretical analyses. Weak condition input increases the difficulties of MLE training. However, ANT gets 0.013 in FED and 31.12 in Inverse BLEU, which are better than other NAR models and close to AR models like ScratchGAN and InitialGAN. It narrows the performance gap between NAR and AR models in tasks with weak conditions. Another GAN-based NAR model, NAGAN, is inferior to all the other models, which shows the limitations of the biased *straight-through estimator*.

For the EMNLP dataset, Transformer is still the best model. ANT outperforms other NAR models in FED, while CMLM can slightly outperform ANT in Inverse BLEU. The iterative decoding mechanism help CMLM to better process complicated datasets with higher decoding latency. To further discuss their performance in token level, we draw the curve of Self-BLEU and Negative BLEU by tuning the temperature in Transformer and show the results in Figure 6 (a). ANT is the only NAR model which can get comparable performance with AR models, while other NAR models (including CMLM) remain behind obviously. Specifically, NAGAN gets extremely low BLEU and Inverse BLEU, which reveals the difficulties of NAGAN to converge on complicated datasets. Furthermore, we compare Least Coverage Rate (LCR) of Transformer and other NAR models in Figure 6 (b). It shows that ANT is the only NAR model which can get close performance with Transformer.

### 4.5.2 Conditional Generation

The experimental results of conditional generation are shown in Table 2. Overall, Transformer gets the best performance in all the evaluation metrics with more decoding iterations. Among NAR models, ANT gets comparable performance with CMLM in FED, and achieves higher Inverse BLEU and Accuracy with lower decoding latency. NAT, which has the same decoding iterations as ANT, is inferior to other models. Fully NAR models trained with MLE rely on informative input representations, so they will meet additional difficulties when adapting to the weak condition tasks.

---

[5]https://www.tensorflow.org

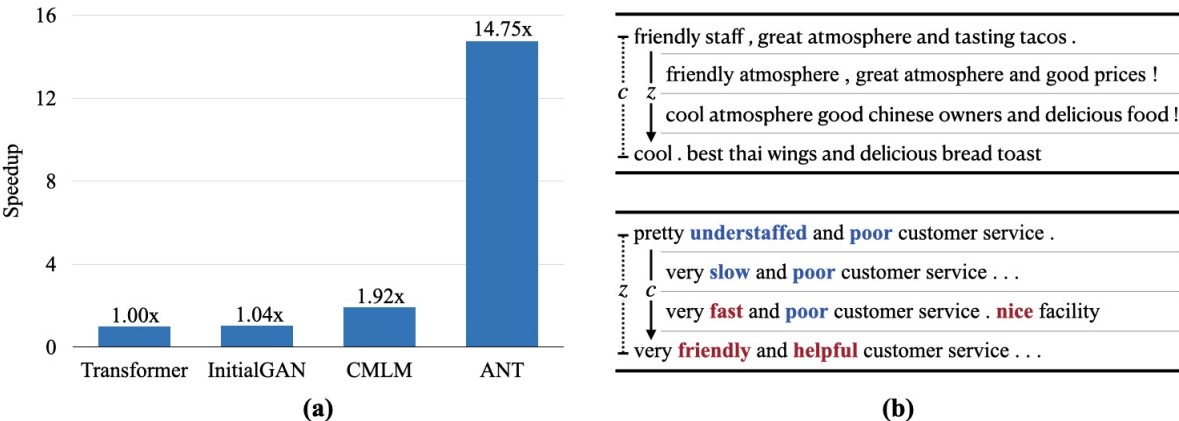

Figure 8: (a) Speedup of different models. ANT achieves the highest speedup, which is 14.75× faster compared to Transformer. (b) Case study of latent interpolation. The generated results can be gradually transformed from one sentence to another by interpolating between latent variables.

### 4.6 Discussion

We conduct ablation studies to evaluate the effectiveness of our proposed components. First, we examine the impact of Dependency FFN on the discriminator's dependency modeling capacity by comparing its performance against the standard FFN, with results presented in Figure 7 (a) and (b). ANT equipped with Dependency FFN achieves lower FED scores, while its BLEU, Self-BLEU, and Inverse BLEU metrics all exhibit closer alignment with the training data distribution. This indicates that Dependency FFN enables ANT to better approximate the target distribution. Notably, when replaced with standard FFN, the model's BLEU score drops substantially, suggesting increased generation of ungrammatical sentences. This degradation demonstrates that standard FFN alone is insufficient for capturing accurate word dependencies in the adversarial training framework.

Furthermore, we investigate the effectiveness of Position-Aware Self-Modulation by analyzing the training convergence behavior on the COCO dataset. As illustrated in Figure 7 (c), ANT with Position-Aware Self-Modulation demonstrates faster convergence and ultimately achieves superior performance compared to the standard self-modulation approach.

One advantage of ANT is that it only requires one decoding step and has high speedup. We compare the speedup of different models in Figure 8 (a). ANT is 14.75 times faster than Transformer. Even comparing with CMLM, it also has much lower decoding latency while obtaining comparable or even better performance. Besides, ANT has great potential in various applications. We explore the potential of ANT in different applications including semi-supervised learning and latent interpolation in the following.

Generative models can also boost the performance of classification models based on semi-supervised learning (SSL). We investigate the application of ANT in SSL by incorporating it into the training of a classification model. The classification model is trained to identify emotion labels of the sentences in the Yelp dataset. We prepare two training sets. One is composed of 500 labeled data and the other one consists of 1,000 labeled data. The results are shown in Table 3. The classification models trained in SSL consistently outperform the ones trained in supervised learning (SL). ANT can help the classification model capture more accurate data distribution so as to achieve better performance.

The potential of ANT can be further explored in controllable text generation. There are two latent variables in ANT: $z$, which is sampled from a pre-defined distribution; and $c$, which is a condition representation. We fix one of them and gradually change the other one. The first case in Figure 8 (b) shows the samples given by tuning $z$, in which ANT transforms one sentence into another one, with the middle sentences kept understandable. The second case in Figure 8 (b) shows the samples given by changing $c$ from the negative representation to the positive representation. ANT gradually transforms negative words into positive ones

Table 3: Effectiveness of ANT in Semi-supervised Learning (Num.: number of labeled data).

| Method | Num. | P | R | F1 |
|--------|------|------|------|------|
| SL | 500 | 91.28% | 89.06% | 90.15% |
| SSL | | 90.77% | 92.15% | **91.46%** |
| SL | 1000 | 92.42% | 91.33% | 91.87% |
| SSL | | 94.87% | 92.39% | **93.62%** |

while keeping the main structure of the sentence. Such controllable generation is seldomly explored by NAR models, and it may inspire further ideas for related tasks.

## 5   Conclusion

In this work, we reveal the limited application scenarios of existing MLE-based fully NAR models due to their incapacity in tasks with weak conditions. We illustrate the reasons from both empirical and theoretical aspects. The inherent convergence problem in MLE-based fully NAR models make it incapable in weak condition tasks. Instead, GANs denote a more promising method. We thus propose an Adversarial Non-autoregressive Transformer (ANT) based on GANs. ANT supports two novel features: Position-Aware Self-Modulation and Dependency FFN. With the help of these two facilities, ANT narrows the performance gaps between fully NAR models and AR models in weak condition tasks with high efficiency. We also demonstrate the potential of ANT in both semi-supervised learning and controllable text generation.

Although ANT can significantly reduce the decoding latency, it still does not outperform Transformer. In the future, we will explore more techniques to further improve its performance. In addition, we will also scale up ANT and apply it to more complicated scenarios.

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

# A Proof

## A.1 The Convergence Problem in Fully NAR Models

Huang et al. (2022a) have theoretically proven the convergence problem of fully NAR models. For clarity and completeness, we reproduce their theoretical proof below.

$$
\begin{aligned}
\min_\theta & KL(P_{data}(Y|X)||P_\theta(Y|X)) \\
&= -H_{data}(Y|X) - \mathbb{E}_{P_{data}(Y|X)}(\sum_{i=1}^{M} \log P_\theta(y_i|X)) \\
&= -H_{data}(Y|X) - \sum_{i=1}^{M} \mathbb{E}_{P_{data}(y_i|X)}(\log P_\theta(y_i|X)) \\
&\geq -H_{data}(Y|X) + \sum_{i=1}^{M} H_{data}(y_i|X)
\end{aligned}
\tag{6}
$$

where $H_{data}(\cdot|X)$ denotes the Shannon entropy, and $C = -H_{data}(Y|X) + \sum_{i=1}^{M} H_{data}(y_i|X)$ is a non-negative constant referred to as the conditional total correlation or multi-information. Further details regarding this theoretical analysis can be found in the original work (Huang et al., 2022a).

## A.2 Proof of Theorem 1.

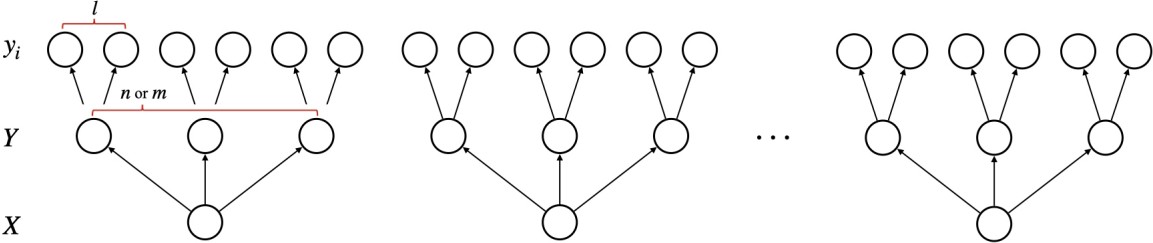

Figure 9: Mapping Relations Described in Assumption 1.

**Assumption 1** describes a relation which is shown in Figure 9. Based on it, we can obtain the following conditional probabilities: $P_{data}(Y|X_A) = P_{data}(y_i|X_A) = 1/n$ and $P_{data}(Y|X_B) = P_{data}(y_i|X_B) = 1/m$. It should be noted that $P_{data}(y_i|X_A)$ and $P_{data}(y_i|X_B)$ are not related to the sequence length $l$ in this scenario. Given a specific input, there are only $n$ or $m$ possible candidate elements at each position. Based on the conditional probabilities, we have:

$$C_A = \sum_{i=1}^{l} H_{data}(y_i|X_A) - H_{data}(Y|X_A)$$
$$= -l \cdot log(1/n) + log(1/n) \tag{7}$$
$$= -(l-1) \cdot log(1/n)$$

$$C_B = \sum_{i=1}^{l} H_{data}(y_i|X_B) - H_{data}(Y|X_B)$$
$$= -l \cdot log(1/m) + log(1/m) \tag{8}$$
$$= -(l-1) \cdot log(1/m)$$

Thus, we can calculate their difference as follows:

$$C_A - C_B = (l-1) \cdot [-log(1/n) + log(1/m)]$$
$$= (l-1) \cdot log(n/m) \tag{9}$$

When $n > m$, $C_A - C_B$ will be positive ($l$ is always higher than 1 in real tasks). It indicates the increase of the conditional total correlation, while higher values indicate higher difficulties in the optimization of MLE-based NAR models. In addition, our experimental results also illustrate the limitations of existing NAR models in tasks with weak conditions, which is consistent with our theoretical analyses.

## B  Experiment Details

### B.1  Implementation of NAT and CMLM

Both NAT (Gu et al., 2018) and CMLM (Ghazvininejad et al., 2019) are designed for machine translation, so their original structures contains an encoder to encode input in source language. However, there may be no meaningful input in our experiment (e.g., unconditional generation), so this structure cannot be used in the experiment directly. We thus use the idea of VAE to obtain hidden representations, so they can be transferred to the tasks in our experiments.

More specifically, a Transformer-based encoder is adopted to encode the sentences into hidden representations during training. Then, these representations are fed into the decoder to reconstruct the input sentences. They use the training objective of VAE, so the representations can be close to the standard normal distribution. During inference, representations sampled from the standard normal distribution will be fed into the decoder, and the decoder will generate sentences based on the sampled representations. For NAT, the representations are fed into decoder as input. For CMLM, the representations are concatenated with the embeddings of input tokens (masked or unmasked words). The iteration number of CMLM is set to be 10 as in previous work (Ghazvininejad et al., 2019; Huang et al., 2022c).

### B.2  Evaluation Metrics

**Fréchet Embedding Distance (FED)** (de Masson d'Autume et al., 2019) is same with the Fréchet Inception Distance (FID) (Heusel et al., 2017) except for the encoding model. The encoding model is changed to be fit into text generation. We adopts Universal Sentence Encoder following the settings of the previous work (de Masson d'Autume et al., 2019). It is calculated as follows:

$$FED = ||\mu_1 - \mu_2||_2^2 + Tr(c_1 + c_2 - 2(c_1 c_2)^{1/2}) \tag{10}$$

where $\mu_1$ and $\mu_2$ are the mean, and $c_1$ and $c_2$ are the covariance.

**Least Coverage Rate (LCR)** (Ren & Li, 2024) is proposed to be a compliment when the FED of two models are too close to each other, since LCR is more sensitive to the change of data quality (Ren & Li,

2024). Given two sets of sentence $X_i^a \in \mathbb{X}_a$ and $X_i^b \in \mathbb{X}_b$, LCR is calculated as follows:

$$
\begin{aligned}
S_{ij} &= Sim(\mathbf{Emb}(X_i^a), \mathbf{Emb}(X_j^b)) \\
R_a &= \frac{1}{|\mathbb{X}_a|} \sum_{i=1}^{|\mathbb{X}_a|} \delta(\sum_{j=1}^{|\mathbb{X}_b|} S_{ij} \geq \tau) \\
R_b &= \frac{1}{|\mathbb{X}_b|} \sum_{j=1}^{|\mathbb{X}_b|} \delta(\sum_{i=1}^{|\mathbb{X}_a|} S_{ij} \geq \tau) \\
LCR(\mathbb{X}_a, \mathbb{X}_b) &= min(R_a, R_b)
\end{aligned}
\tag{11}
$$

where $X_i^a$ and $X_i^a$ are the i-th and j-th sentences from sentence sets $\mathbb{X}_a$ and $\mathbb{X}_b$), respectively. $\mathbf{Emb}(\cdot)$ is the model used to transform sentences into embeddings (which is Universal Sentence Encoder in this work), $\tau$ is a hyperparameter, $Sim(\cdot)$ is cosine similarity and $\delta(\cdot)$ is a function which returns 1 if input is higher than 0, and 0 for others.

The idea of LCR is to identify whether a specific mode in one set is covered by the sentences in another set or not. Then, it uses the minimum coverage rates as the output, so LCR can be sensitive to two common problems in text generative models: 1) models tend to generate sentences which are out of the real distributions; and 2) the generated sentences are in high similarities.

All the token level metrics (i.e., **BLEU**, **Self-BLEU**, and **Inverse BLEU**) are calculated up to 5 grams following previous work (de Masson d'Autume et al., 2019; Ren & Li, 2024).

