# OpenReview forum: "Unlocking the Power of GANs in Non-Autoregressive Text Generation under Weak Conditions"
_TMLR — Rejected by TMLR_

### Review · Reviewer_ox3p · 2025-07-28

**Summary Of Contributions:**

The paper proposes an adversarial non-autoregressive transformer for weakly conditioned language tasks. The authors theoretically analyze the limitations of existing non-autoregressive transformers and introduce a GAN-based two-stage training framework to address them. First, a mapper is trained to encode token representations; then, a generator and discriminator are jointly trained for text generation. To enhance performance, the authors introduce position-aware self-modulation to improve representation diversity and a Dependency Feed-Forward Network to better model word dependencies. They also extend the method to conditional generation. Experiments on the COCO Image Caption and EMNLP 2017 News datasets show competitive results against several GAN-based baselines. Ablation studies confirm the effectiveness of the proposed components. Additionally, the model shows promise in applications such as latent space interpolation and semi-supervised learning.

**Audience:**

Yes

**Claims And Evidence:**

Yes

**Requested Changes:**

The paper is overall solid, and I have no major change requests. Below are a few minor issues I noticed:

1. In Assumption 1: "Each $Y$ can only by mapped by a specific input." $\rightarrow$ should be "Each $Y$ can only be mapped by a specific input."

2. The word “exiting” appears multiple times and should be corrected to “existing.”

3. On page 2, “which provide” $\rightarrow$ “which provides”.

4. Consider adding more description to the caption of Figure 2 to improve clarity.

**Strengths And Weaknesses:**

Strengths:

1. The proposed method is novel in applying GANs to non-autoregressive transformers.

2. The authors demonstrate that it achieves faster inference than traditional transformers.

3. Comprehensive experiments and ablation studies support the effectiveness of the method.

4. The approach shows potential for latent space interpolation and semi-supervised learning.

Weaknesses:

1. The proposed method underperforms compared to standard transformers.

---

> ### Author Response · Authors · 2025-08-18
> **Reply to Reviewer ox3p**
>
> Thank you for your comments. Please find our responses to your concerns below:
>
> Regarding Weakness 1: While our model does not outperform standard Transformers, it has significantly narrowed the performance gap between fully NAR models and AR models in weak condition tasks. As a pioneering study exploring the application of GANs in constructing NAR models, this work aims to demonstrate the effectiveness and potential of this approach. We will further improve model performance in our future work.
>
> Regarding the typographical errors: We are sorry about the typos. In addition to correcting the specific typos you identified, we have conducted a comprehensive review of the manuscript to identify and rectify the typos we can find throughout the paper.
>
> Regarding the figure captions: We have revised the figure captions throughout the paper to provide more detailed explanations and improved clarity.
>
> We have updated a new version of the manuscript incorporating all these revisions for your reference. Please do not hesitate to let us know if you have any additional concerns.

---

### Review · Reviewer_zc1E · 2025-08-11

**Summary Of Contributions:**

The paper proposed a GAN model for non-autoregressive text generation (although some part of the architecture is still autoregressive, i.e., dependency FFN) and demonstrated it achieves comparable performance with mainstream NAR methods in tasks with weak conditioning.

**Audience:**

No

**Broader Impact Concerns:**

This is a methodology paper and there is no obvious negative social impact.

**Claims And Evidence:**

No

**Requested Changes:**

### Minor

* Assumption 1: "is consisted of" -> consists of
* Eq. (1): MSA -> MHA

**Strengths And Weaknesses:**

Overall, I would say this paper requires substantial revision before it can be properly reviewed. The manuscript in its current form is hindered by numerous typos, undefined mathematical notations, and unclear descriptions of the methodology.

* The core motivation for this work is questionable. The paper claims that for Non-Autoregressive (NAR) methods, "the learned distributions cannot be identical to the target distributions unless all the words in a sentence are independent of each other" - I believe this is a False claim since modern NAR methods, such as those based on masked diffusion, are specifically designed to effectively model the joint distribution of sequences.

* Key mathematical notations are used without proper definition. Is the definition of C correct in section 2.1.1? What are H_data(y_i|X) and H_data(Y|X) exactly? Can you define them? I find the description of Assumption 1 hard to understand - what do you mean by "each Y can only by mapped by a specific input"?

* The training objective is confusing. In fact I find the components conflicting with each other. E.g., it was mentioned that the mapper network is trained as a BERT but then in eq. (2) another objective based on VAE is introduced, again without definition of the p(x|z) distribution and how it relates to the network. In fact, reading this section gives me the feeling that the paper is AI generated.

* The methodology sections lack evidence - For example, it's said "Transformer (Vaswani et al., 2017) builds word dependencies by dynamically assigning weights in the attention mechanism. This process, however, is unstable under the training of GANs. It will lead the models to lose word dependencies, and finally result in ungrammatical sentences" - This argument is not supported with any evidence - it is also not sensible.

* Experiments lack strong NAR baselines such as masked diffusion that are current SoTAs for unconditional (weak conditioned) generation.

---

> ### Author Response · Authors · 2025-08-18
> **Reply to Reviewer zc1E**
>
> Thank you for your reviews. We provide our detailed responses to your concerns below:
>
> Regarding Weakness 1: We apologize for any confusion in our original presentation. This statement specifically pertains to fully non-autoregressive (NAR) models, which employ only a single decoding step to generate complete outputs. In contrast, diffusion models require a number of iterative steps to produce samples, and therefore do not qualify as fully NAR models. While fully NAR models offer the highest computational speedup, their performance typically lags behind autoregressive (AR) models in weak condition tasks. This work addresses this fundamental limitation by leveraging GANs to train fully NAR models effectively. Our experimental results demonstrate both remarkable performance and significant speedup of the proposed models. We have revised the illustration in the introduction's first paragraph to avoid potential ambiguity.
>
> Regarding Weakness 2: The theoretical analysis presented in the first paragraph is from previous work by Huang et al. [1], where H_{data} represents the Shannon Entropy. To enhance comprehension, we have included the complete mathematical proof from Huang et al. [1] in Appendix A.1 of our revised version. Regarding the typographical error in Assumption 1, we sincerely apologize for this oversight. The correct statement should read: "Each Y can only be mapped by a specific input." These relationships are also illustrated in Figure 9 of Appendix A.2. We have corrected this error to improve the clarity and accuracy of this section.
>
> Regarding Weakness 3: We apologize for any confusion in our original explanation. During the training of the mapper, we randomly mask a specified number of tokens and task the model with predicting these masked tokens (similar to BERT's approach), while the training objective employs the VAE loss function. Specifically, q(z|x) describes the encoding process that transforms words into latent representations z, while p(x|z) represents the decoding process that recovers latent representations back into specific words. P(z) is assumed to follow a standard normal distribution, consistent with standard VAE methodology. We have revised section 3.1 to enhance the clarity of this component.
>
> Regarding Weakness 4: We have provided additional comprehensive results addressing this concern in our revised version. As demonstrated in Figure 7(b), the BLEU score, which measures sentence fluency, decreases significantly when using standard FFN. This indicates that ANT with conventional FFN generates less fluent sentences, thereby confirming the presence of inaccurate word dependencies captured by the model. We have enhanced Figure 7 and expanded the first paragraph of section 4.6 to provide more detailed explanations of these findings.
>
>
> Regarding Weakness 5: Existing diffusion models for text generation require numerous iterative steps to produce samples [2-4], with many approaches being slower than standard autoregressive Transformers [3]. The primary focus of this work is to develop a fully NAR model that achieves superior computational speedup. As a pioneering study investigating fully NAR models for challenging tasks, this work aims to demonstrate the effectiveness and potential of this methodology. While our model achieves remarkable improvements among fully NAR models in weak condition tasks, we acknowledge that it does not yet outperform models that require significantly more decoding steps. We plan to continue advancing model performance to bridge this gap while preserving the computational efficiency advantages.
>
> Regarding the requested changes: We sincerely apologize for the typographical errors in our original submission. We have conducted a comprehensive review of the entire manuscript and corrected all identified errors to improve readability and precision.
>
> We have prepared an updated version of the manuscript incorporating all these revisions for your consideration. Please do not hesitate to contact us if you have any additional concerns or require further clarification.
>
>
>
>
> [1] Huang, Fei, et al. "On the learning of non-autoregressive transformers." International conference on machine learning. PMLR, 2022.
>
> [2] Li, Xiang, et al. "Diffusion-lm improves controllable text generation." Advances in neural information processing systems 35 (2022): 4328-4343.
>
> [3] Sahoo, Subham, et al. "Simple and effective masked diffusion language models." Advances in Neural Information Processing Systems 37 (2024): 130136-130184.
>
> [4] Nie, Shen, et al. "Large language diffusion models." arXiv preprint arXiv:2502.09992 (2025).

---

### Review · Reviewer_MGbp · 2025-08-11

**Summary Of Contributions:**

This paper proposes an Adversarial Non-autoregressive Transformer (ANT) for weak condition tasks (i.e., unconditioned or conditioned generation). The proposed approach consists of three modules: mapper, generator, and discriminator. The mapper converts words
into representations, and the generator tries to recover these representations. The discriminator needs to identify whether input representations are from the mapper or the generator. Moreover, this work also develops theories to analyze the limitation of MLE-based NAR approaches.

**Audience:**

Yes

**Claims And Evidence:**

Yes

**Requested Changes:**

- The authors should point out more connections between the theories and practical methods.
- It is unclear what losses are used to train the mapper. First, it mentions *the mapper is trained to reconstruct words based on the masked input similar to BERT* and also *we use the loss function of variational autoencoder (VAE) (Kingma & Welling, 2014) to train the mapper in Eq. (2)*. The authors need to write the total loss in a clearer form.
-  It is unclear about the inputs to the generator (i.e., trainable representations). As far as I guess, they are trainable representations learned during training. Moreover, Section 3.2 about Position-Aware Self-Modulation is hard to comprehend with the appearance of unexplained symbols such as $z$ and $x_i$ in Eq. (3).
  - The authors should rewrite Section 3.2 to make it clearer and easy to comprehend.
  - The authors should improve Figure 2 on Position-Aware Self-Modulation and Dependency FFN.
  - The caption of Figure needs to be more informative.

**Strengths And Weaknesses:**

## Strengths
- The theories make sense because it demonstrates that more complex datasets lead to higher total correlations for NAR.
- The approach proposes some novel components such as 1) Position-Aware Self-modulation which provides more effective input signals to assist the model to obtain diverse words in a sentence; and 2) Dependency Feed Forward Network (Dependency FFN) which further improves model performance by enhancing its capacity in dependency modeling.
- The experimental results are good and the proposed approach benefits in the decoding time.

## Weaknesses
- The paper could benefit more from a clearer and better writing. Some technical parts lack descriptions to comprehend fully.
- The novelty is somehow limited because it leverages existing techniques with appropriate modifications.
- There is no connection between the theories and practical methods.

---

> ### Author Response · Authors · 2025-08-18
> **Reply to Reviewer MGbp**
>
> Thank you for your comments. We provide our responses to your concerns below:
>
> Regarding Weakness 1: We have substantially revised sections 3.2 and 3.3 to enhance clarity and comprehensiveness. Additionally, we have incorporated two new figures to better illustrate our methodology: Figure 4 presents a detailed comparison between self-modulation and position-aware self-modulation, while Figure 5 demonstrates the differences between standard feed-forward networks and our proposed dependency feed-forward network. We hope that these visual aids, combined with our expanded explanations, can make our technical parts clearer.
>
> Regarding Weakness 2: The novelty of this paper can be demonstrated from several key aspects. First, we identify the fundamental limitations of MLE in NAR text generation and propose a more promising approach for training fully NAR models. Furthermore, to ensure that ANT (our proposed model) achieves remarkable performance, we propose and incorporate two novel components: (1) Position-Aware Self-modulation, which generates effective signals enabling the model to produce diverse words within a sentence; and (2) Dependency Feed-Forward Network (Dependency FFN), which helps the model capture more accurate word dependencies during the inherently unstable training process of GANs. Our experimental results demonstrate the effectiveness of these proposed techniques. To the best of our knowledge, this is the first work to demonstrate the effectiveness of GANs in NAR models. Our proposed techniques also play a crucial role in the remarkable performance of ANT.
>
> Regarding Weakness 3 and Requested Change 1: Our theoretical analysis focuses on demonstrating the limitations of MLE-based NAR models in weak condition tasks. However, GANs do not suffer from such convergence issues, and their convergence properties have been rigorously established in prior literature. Therefore, we construct our models based on GANs rather than MLE. In the revised version, we have added detailed explanations regarding this design choice at the end of section 2.1.1.
>
> Regarding Requested Change 2: We apologize for any confusion in our original presentation. During the training of the mapper, we randomly mask a specified number of words and task the model with predicting these masked tokens (similar to BERT's approach), but the training objective utilizes the loss function in VAE. This methodology establishes a continuous representation space for each word, ensuring that the representations generated by our model can be accurately decoded to the correct words even when they deviate slightly from their optimal points. We have revised the relevant explanations in section 3.1 to enhance clarity.
>
> Regarding Requested Changes 3 and 4: We employ a trainable matrix as the initialization point and integrate latent variables through our proposed position-aware self-modulation mechanism. Subsequently, these results are processed through transformer blocks to obtain the final word representations. We have thoroughly revised section 3.2 and incorporated additional detailed explanations to improve the clarity of this component.
>
> Regarding Requested Change 5: We have prepared new figures for each component. Specifically, we present a detailed comparison between self-modulation and position-aware self-modulation in Figure 4, and illustrate the architectural differences between standard FFN and our dependency FFN in Figure 5. We believe these visual representations will enhance the understanding of our methodology.
>
> Regarding Requested Change 6: We have incorporated additional detailed explanations for each figure, enabling readers to comprehend these figures more intuitively and effectively.
>
> We have updated a new version of the manuscript incorporating all these revisions for your reference. Please do not hesitate to let us know if you have any additional concerns.

---

### Decision · Action_Editor_H1Lz · 2025-10-09

**Recommendation:** Reject

**Additional Comments:**

Given disagreements between reviewers, I went ahead and carefully read the submission. I find it has merits in terms of interest to TMLR's readership, but suffers from issues regarding clarity and the validity of several claims that outweigh those merits. I would encourage the authors to address the issues outlined in the Claims and evidence section of the decision and resubmit a revised version of the manuscript.

**Audience:**

Yes

**Audience Explanation:**

The reviewers' majority opinion is that the submission is of interest to at least some individuals in TMLR's audience:

* "The approach proposes some novel components such as 1) Position-Aware Self-modulation which provides more effective input signals to assist the model to obtain diverse words in a sentence; and 2) Dependency Feed Forward Network (Dependency FFN) which further improves model performance by enhancing its capacity in dependency modeling." (MGbp)
* "The proposed method is novel in applying GANs to non-autoregressive transformers." (ox3p)
* "The approach shows potential for latent space interpolation and semi-supervised learning." (ox3p)

Reviewer MGbp finds that the theory presented is only loosely connected to the empirical results, to which the authors respond that their "theoretical analysis focuses on demonstrating the limitations of MLE-based NAR models in weak condition tasks". Having looked at the theoretical results, they appear to be a somewhat straightforward application of Huang et al.'s result to the case of a uniform distribution over "possible candidate elements", and I can see the connection to the "weak condition tasks" the submission investigates.

Overall, I agree with the majority opinion on the Audience criterion: fully non-autoregressive approaches to text generation are a worthwhile pursuit, and the submission proposes interesting ideas in that area.

**Claims And Evidence:**

No

**Claims Explanation:**

In terms of evidence, Reviewer MGbp finds the submission presents good experimental results, and Reviewer ox3p finds that "comprehensive experiments and ablation studies support the effectiveness of the method".

Reviewer zc1E however notes that experiments lack strong NAR baselines such as masked diffusion that are current SoTAs for unconditional (weak conditioned) generation. The authors counter that "existing diffusion models for text generation require numerous iterative steps to produce samples", and that the "primary focus of [their] work is to develop a fully NAR model that achieves superior computational speedup". The abstract however goes a step further than that in claiming that "experimental results demonstrate that ANT achieves comparable performance to mainstream models while maintaining significantly higher efficiency"; this raises the question of whether text diffusion models are considered "mainstream models". The authors should either nuance the claim in the abstract or broaden the scope of compared approaches.

The main reviewer concerns center around clarity and the correctness of statements made in the main text. Both Reviewers MGbp and zc1E find that the paper lacks sufficient details on the proposed approach:

* "some technical parts lack descriptions to comprehend fully" (MGbp)
* "hindered by numerous typos, undefined mathematical notations, and unclear descriptions of the methodology" (zc1E)

The authors made a valiant effort to clarify in their response and in the updated manuscript, but to Reviewer zc1E (and, after having read the submission carefully, to me as well) this is not enough to fully address those concerns.

* Both Reviewer MGbp and Reviewer zc1E are confused by what loss is used to train the mapper. In their response, the authors clarified the nature of the loss and updated the explanations in Section 3.1, but to Reviewer zc1E (and to me) this is still not clear enough. Reviewer zc1E writes: "Is $x$ the clean or masked tokens? How is VAE training loss supposed to predict the masked tokens?" I would also add that in its current form, the submission only specifies that "the mapper is trained to reconstruct words based on the masked input", but no indication is given in Figure 2 or Equations 1 and 2. Furthermore, the notation $q(z_i' | x_i)$ suggests that the latent variable $z_i'$ only depends on $x_i$, whereas MHA would introduce a dependency on all other words in the sentence if I understand correctly.
* Reviewer zc1E finds that the following claim is "not supported with any evidence": "Transformer (Vaswani et al., 2017) builds word dependencies by dynamically assigning weights in the attention mechanism. This process, however, is unstable under the training of GANs. It will lead the models to lose word dependencies, and finally result in ungrammatical sentences." The authors provide additional results in Figure 7(b) and claim that the results indicate that "ANT with conventional FFN generates less fluent sentences, thereby confirming the presence of inaccurate word dependencies captured by the model". However, the main text still makes the following claim that is not backed up by evidence: "During adversarial training, the competing objectives between generator and discriminator can interfere with the learning of attention patterns, making it difficult for the model to consistently capture syntactic and semantic dependencies between words. [...] To address this limitation, we propose the Dependency Feed Forward Network (Dependency FFN), which explicitly strengthens the traditional FFN module with enhanced capacity for dependency modeling." The experiment in Figure 7(b) shows that Dependency FFN indeed improves performance over FFN, but it does not demonstrate that the improvement is due specifically to the enhanced capacity for dependency modeling and not, e.g., the additional capacity afforded by the weight matrix it introduces in Linear 3. I don't think the ablation design is sufficient to support such a claim.

In the course of reading the updated manuscript and author response, I also noted the following incorrect claims and clarity issues:

* "Unlike MLE, which is fundamentally incompatible with NAR models [...]" This ("fundamentally incompatible") is a strong statement. Would energy-based models not qualify as principled, MLE-based NAR approaches?
* "The synthetic distributions learned by GANs can theoretically converge to the real distributions, ensuring their convergence even under weak conditions. [...] Unlike MLE-based NAR models whose convergence cannot be guaranteed and is easily influenced by input conditions, the convergence of GANs has been proven regardless of input conditions Goodfellow et al. (2014). [(in the authors' response:)] However, GANs do not suffer from such convergence issues, and their convergence properties have been rigorously established in prior literature." This overclaims the convergence of GANs, which I believe stems from conflating "convergence" with "global optimality". Goodfellow et al. (2014) demonstrate that the generator matching the data distribution is the globally optimal solution to generative adversarial training, but their proof of convergence only holds provided the generator and discriminator have enough capacity and at each step of the algorithm the discriminator is allowed to reach its optimum. In practice, neither assumptions hold and convergence is not guaranteed. See for instance Gidel et al. (2018) for a discussion of convergence in adversarial training. This also overlooks mode collapse issues in GANs; see for instance Bau et al. (2019).
* Symbols not introduced in the text: in Equation 1, is $h_0$ a token, or a token embedding?
* Overloaded notation: variables $x$ and $h$ are reused throughout, sometimes to represent the same thing under different notations ($h_0$ in Equation 1 and $x$ in Equation 2), and sometimes to represent different things using the same notation ($h$ in Equations 1 and 3, $x$ in Equations 2, 3, and 4).
* Figure 3 does not specify how the plots were obtained, what data was used to train the model from which the plots were obtained, etc.
* It's unclear what dataset was used for Figure 7(a,b).
* One of the compared approaches is simply named "Transformer"; what specific architecture was used? Is it comparable in capacity to ANT? How was the 14.75x speedup computed when comparing ANT and Transformer?

References:

* Gidel, G., Berard, H., Vignoud, G., Vincent, P., & Lacoste-Julien, S. (2018). A variational inequality perspective on generative adversarial networks. arXiv preprint arXiv:1802.10551.
* Bau, D., Zhu, J. Y., Wulff, J., Peebles, W., Strobelt, H., Zhou, B., & Torralba, A. (2019). Seeing what a GAN cannot generate. In Proceedings of the IEEE/CVF international conference on computer vision (pp. 4502-4511).

**Resubmission Of Major Revision:**

The authors may consider submitting a major revision at a later time.